# Are childbirth location and mode of delivery associated with favorable early breastfeeding practices in hard to reach areas of Bangladesh?

Nazia Binte Ali[1]*, Farhana Karim[1], S. K. Masum Billah[1], Dewan M. D. Emdadul Hoque[2], Abdullah Nurus Salam Khan[1,3], Mohammad Mehedi Hasan[1], Sonjida Mesket Simi[1], Shams E. L. Arifeen[1], Mohiuddin Ahsanul Kabir Chowdhury[1,3]

1 Maternal and Child Health Division, International Centre for Diarrhoeal Disease Research, Bangladesh (icddr,b), Dhaka, Bangladesh, 2 United Nations Population Fund, Dhaka, Bangladesh, 3 Department of Epidemiology and Biostatistics, Arnold School of Public Health, University of South Carolina, Columbia, South Carolina, United States of America

* nazia.ali@icddrb.org

**Editor:** Julia Dratva, Zurich University of applied sciences, School of Health Professions, Leitung Forschunsstelle Gesundheitswissenschaften/Head of health sciences research, SWITZERLAND

## Abstract

### Background

Early initiation of breastfeeding within one hour of birth (EIBF) and no prelacteal feeding are WHO recommended practices for improving maternal and newborn health outcomes. Globally, EIBF can avert around 22% of newborn death. In recent years, Bangladesh has experienced increasing facility delivery coverage and cesarean section rates. However, the impact of these changes on early breastfeeding initiation in hard to reach areas (HtR) of the country is still poorly understood. Therefore, this study aimed to examine the independent associations between childbirth locations and mode of delivery with favorable early breastfeeding practices in four hard to reach areas of Bangladesh.

### Method

We extracted data from a cross-sectional study conducted in four HtR areas of Bangladesh in 2017. A total of 2768 women, having birth outcomes in the past 12 months of the survey, were interviewed using structured questionnaires. EIBF and no prelacteal feeding were considered as favorable early breastfeeding practices. The categories of childbirth locations were defined by the place of birth (home vs. facility) and the delivery sector (public/NGO vs. private). The mode of delivery was categorized into vaginal delivery and cesarean section. Generalized linear models were used to test the independent associations while adjusting for potential confounders.

### Results

The prevalence of EIBF practices were 69.6%(95% CI:67.8–71.3); 72.2%(95% CI:67.8–71.3) among home births Vs 63.0%(95% CI:59.5%-66.4%) among facility births. Around 73.9% (95% CI:72.3–75.6) mother's in the study areas reported no-prelacteal feeding.

**Data Availability Statement:** Due to ethical restrictions related to protecting study participants privacy and confidentiality, data access is restricted by the Ethical Review Committee of icddr,b. According to the icddr,b data policy (http://www.icddrb.org/policies), interested parties may contact Ms. Armana Ahmed (aahmed@icddrb.org) with further inquiries related to data access

**Funding:** The research study was funded by the Swedish International Development Cooperation Agency (Sida) through icddr,b (Grant number: GR 01455). MAKC received funding for this study. URL of funder website: https://www.sida.se/English/. The funders had no role in study design, data collection and analysis, decision to publish, or preparation of the manuscript.

**Competing interests:** The authors have declared that no competing interests exist.

Compared to home births, women delivering in the facilities had lower adjusted odds of EIBF (aOR = 0.51; 95%CI:0.35–0.75). Cesarean section was found to be negatively associated with EIBF (aOR = 0.20; 95%CI:0.12–0.35), after adjusting for potential confounders. We could not find any significant associations between the place of birth and mode of delivery with no prelacteal feeding.

## Discussions

This study found that facility births and cesarean deliveries were negatively associated with EIBF. Although the implementation of "Baby-Friendly Hospital Initiatives" could be a potential solution for improving EIBF and no prelacteal feeding practices, the challenges of reduced service availability and accessibility in HtR areas must be considered while devising effective intervention strategies. Future studies can explore potential interventions to promote early breastfeeding for facility births and cesarean deliveries in HtR areas.

## Background

The benefits of breastfeeding on maternal and child health are well established. Breastmilk is the only natural food for children sufficient to meet all nutritional requirements until six months of age. Colostrum, the golden milk, has high immunogenic properties and protects against infectious diseases such as diarrhea [1], pneumonia, meningitis, and neonatal sepsis [2]. Breastfeeding reduces neonatal and infant mortality by addressing major causes of neonatal and infant deaths and improves nutritional status and cognitive development of children [1–4]. Additionally, the potential benefits of breastfeeding includes birth spacing, lower risk of developing postpartum hemorrhage, breast cancer, ovarian cancer, and type 2 diabetics [5, 6]. The Lancet breastfeeding series 2016 recognizes optimal breastfeeding as a "smart investment" for a nation due to the economic gain achieved by reducing health costs from morbidity and increased productivity of the breastfed children in adult life [6, 7]. Thus, optimal breastfeeding plays a crucial role in achieving several Sustainable Development Goals, such as goals 2, 3, and 4 [8].

WHO and UNICEF recommend early initiation of breastfeeding within one hour of birth (EIBF) and exclusive breastfeeding until six months of age [9, 10]. EIBF prevents hypothermia among newborns through the skin to skin contact, improves the bonding between mother and newborn, and increases the potential for exclusive breastfeeding [11]. Evidence suggests EIBF can avert approximately 22% of neonatal deaths worldwide [3, 4, 12, 13]. On the contrary, delayed breastfeeding initiation is one of the predictors of prelacteal feeding, which is the practice of providing any food or liquid to the newborn other than breast milk in the first three days of life [14]. Prelacteal foods are less nutritious than breastmilk and increase the risk of infection through contamination. Additionally, prelacteal feeding reduces the success rate of exclusive breastfeeding [15, 16].

EIBF and no prelacteal feeding can be predicted by the place of birth and mode of delivery. Although several past studies reported a positive association between facility delivery and EIBF, a negative association is also evident [5, 11, 17–19]. Furthermore, a significant negative association was found between cesarean section deliveries and EIBF practices in Bangladesh [19]. Results suggest that the directions of these associations (positive or negative) could vary depending on the country and study context.

Despite previous attempts to understand the determinants of EIBF in Bangladesh, a study in the context of hard to reach (HtR) areas is still missing [17, 19, 20]. Such a study is important because of multiple reasons. First, while comparing the trends from Bangladesh Demographic and Health Survey (BDHS) in 2014 and 2017, we observed some major shifts in the coverage of facility deliveries (37% vs. 50%), deliveries at private sector (22% vs. 32%) and cesarean section rates (23% vs. 33%) [21, 22]. Similar improvements were also observed in the prevalence of EIBF, which increased from 51% to 69% between 2014 and 2017 [21, 22]. Due to these recent changes, it is essential to reexamine the associations between the place of birth and mode of delivery with EIBF and no preleacteal feeding practices. Second, these changes highlight the importance of employing more recent data from BDHS 2017 to study the determinants of EIBF in HtR areas. It is necessary to understand whether the improvements in facility delivery coverage had any tangible impact on EIBF practices in these areas. Third, a specific focus on HtR areas is crucial as it contains marginalized populations with inadequate access to health care and experiences high child mortality rates [23]. Findings from this study are expected to generate novel evidence on how the place of birth and mode of delivery is related to EIBF and no prelacteal feeding practices in the HtR areas.

To address the existing research gap, this study aimed to determine the relationship between place of birth, mode and sector of delivery with favorable early breastfeeding practices in four hard to reach areas of Bangladesh.

## Method

### Data & study population

A cross-sectional study was conducted by the Maternal and Child Health Division of International Center for Diarrheal Disease Research, Bangladesh (icddr,b) in four hard to reach areas (Char, Hilly, Haor, and Coastal) of Bangladesh in 2017. The primary objective of the study was to identify gaps in key maternal and newborn health (MNH) service delivery in HtR areas using the Tanahashi model [24]. The study used a mixed-method approach. Household survey, health facility assessment survey, and qualitative explorations were conducted for collecting data. In this study, we used data from the household survey.

A total of 2768 recently delivered women who had a birth outcome in the past 12 months of the survey were interviewed. A structured questionnaire, adapted from BDHS 2014, was used to collect data on the background of women and their husbands, socioeconomic status of the households, antenatal and postnatal care, delivery, birth outcome, early initiation of breastfeeding and prelacteal feeding practices.

In this study, we carried out a secondary analysis of the household survey data to explore the association between favorable early breastfeeding practices with the place of birth and mode of deliveries. The final analysis was conducted on data collected from 2721 women, who had live birth outcomes in the past 12 months of the survey and had complete information on the initiation of breastfeeding and prelacteal feeding practices. The participant flow diagram is detailed in Fig 1.

### Sampling

The study used a multistage stratified random sampling technique for the selection of clusters. In the first stage, Satkhira (Coastal), Chittagong (Hilly), Sunamganj (Haor), and Kurigram (Char) districts were selected randomly from the lists of four types of HtR areas. The Government of Bangladesh (GoB) determines these lists of HtR areas and uses six indicators for defining HtR sub-districts; land type, availability of water, access to pure drinking water, sanitation facilities, poverty level, and child mortality rate [23]. Later, three *unions* (smaller units of a

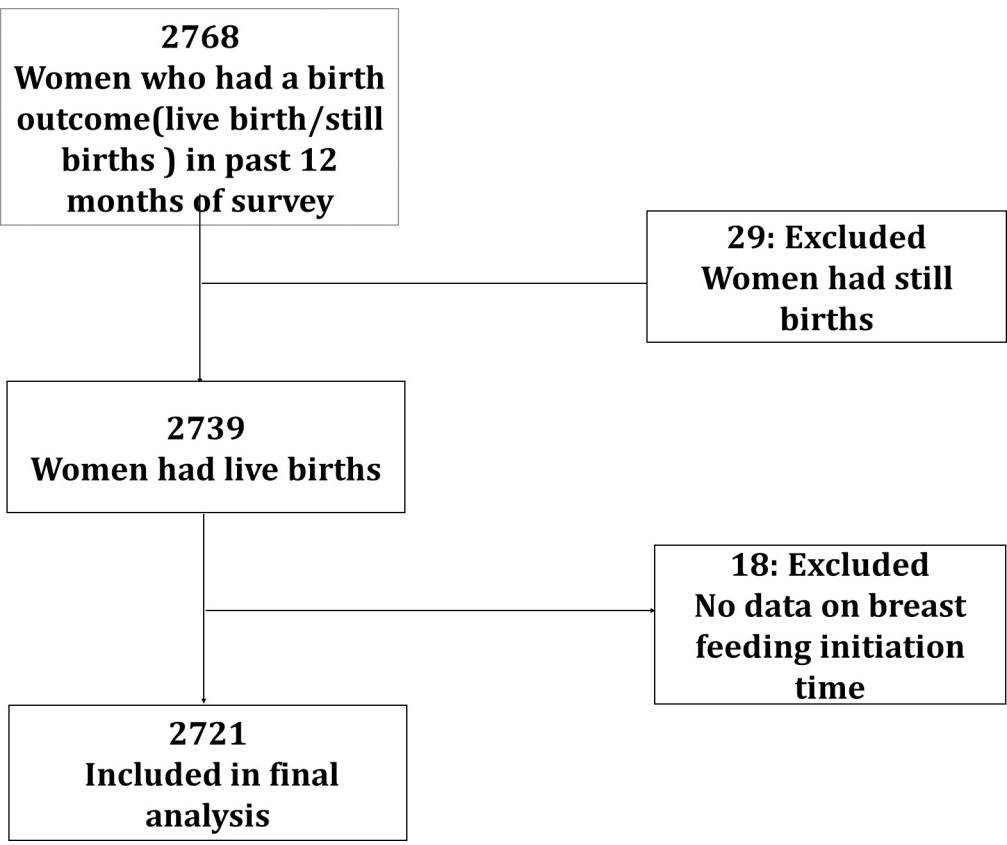

**Fig 1. Participant flow diagram for final analysis.**

sub-district) were randomly selected from each type of HtR area and three clusters were selected randomly from each union. Each village with approximately 1000 population was considered as a cluster. We identified and listed the recently delivered women with a birth outcome in the past 12 months of the survey. All listed women who were present during the interview day and provided informed consent were interviewed.

## Definition of variables

**Dependent variables.** In this study we explored, two dependent variables. Our primary outcome measures were favorable early breastfeeding practices, described by early initiation of breastfeeding (EIBF) and no prelacteal feeding. EIBF is the practice of babies starting breastfeeding with one hour of birth and is a WHO-recommended indicator for assessing Infant and Young Child Feeding (IYCF) practices. The second outcome, no prelacteal feeding, conforms to the scenario where a newborn receives only breast milk in the first three days of life. We used self-reported data from the mothers for defining the exposure and outcome variables. EIBF and no prelacteal feeding data was recorded in binary (yes/no) form.

**Independent variables.** The childbirth location was described by the place of birth and delivery sector. Any birth taking place within a health facility managed by the public, private or non-government organization (NGO) was defined as a facility birth. In contrast, deliveries conducted at home were labeled as home birth. The facility births were further categorized into public and private sector deliveries. NGOs are non-profit organizations and were considered to be similar to the public sector during the modeling. Thus, the births at public and

NGO managed facilities were labeled as public/NGO sector deliveries. The home births was excluded while analyzing the association between the delivery sector and favorable early breastfeeding practices. Delivery by medically trained provider was a binary variable (yes and no). Any delivery conducted by medical doctor, nurse/midwives/paramedic, family welfare visitor, community skilled birth attendent and sub-assistant community medical officer was defined as delivery by medically trained provider.

**Potential confounding variables.** We identified potentiation confounders on a priori basis through a literature review and tested for potential confounding effects before inclusion in the final multivariable model. The potential confounders adjusted in the models are maternal age groups (15–20, 21–30 and 31–49 years), maternal education (no formal education, primary, secondary and above), current employment status (yes or no), religion (Muslim; Hindu and others), household wealth quintile (lowest, second, middle, fourth and highest), number of ANC visits (No ANC, 1, 2, 3, $\geq 4$), complications during pregnancy (yes or no) and complications during delivery (yes or no).

We estimated the household wealth index using the Principle Component Analysis (PCA). A composite score was generated from the household's ownership of selected assets at the time of the survey. The ownership of assets included the possession of electricity, radio, televisions, refrigerators, mobile phone, vehicles (bicycles, motorcycle, rickshaw or boat), furniture, electricity generator, computer or laptop, ownership of home and livestock, construction materials of house, access to drinking water and sanitation facilities. The composite score was then divided into five categories to generate household wealth quintiles.

For defining complications during pregnancy, we used a self-reported history of severe headache, blurring of vision, convulsion, loss of consciousness, high blood pressure, severe bleeding, leaking membrane, no labor pain for six hours and edema of the face or legs during pregnancy. Similarly, complications during delivery were defined by the self-reported history of severe headache with blurred vision, convulsion or fit, high blood pressure, severe bleeding, prolonged labor (>12 hours), retained placenta, edema, and malpresentation of fetus during delivery.

## Analysis

Initially, we conducted a descriptive analysis of all the variables, including the exposure and outcome variables. Findings from the descriptive analysis were reported using weighted proportions. For pooled analysis, weight was calculated from sample weights for each type of HtR area and the population weight of the HtR areas. Bivariate analysis was carried out using generalized linear models (GLM) to identify the associations between exposures and outcome variables [25]. Bivariate models were adjusted for the type of HtR areas, weight and clustering effects. The statistically significant bivariate associations were later tested using multivariable models. Any variable showing significant associations with exposure and/or outcome variables were selected as confounders and considered for inclusion in the final multivariable models. Multivariable models were adjusted for potential confounders in a stepwise approach. First, sociodemographic confounders (maternal age group, education, occupation, and wealth index) were adjusted, followed by a step by step inclusion of the statistically significant variables found in bivariate analyses.

We tested six multivariable models, considering the three exposure variables for each of the two primary outcomes (Table 1). Generalized linear models (GLM) for the binomial family were used for multivariable analysis [25]. GLM uses Newton–Raphson (maximum likelihood) optimization for model fit. We specified the *vce* function of GLM for adjusting clustering among the samples. Weight function was also specified during multivariable modeling. Model

**Table 1. Description of final generalized linear models (GLMs).**

| Model | Dependent variable | Independent variable | Confounders |
|---|---|---|---|
| Model 1 | Early initiation of breastfeeding (EIBF) | Facility births (reference home delivery) | Adjusted for maternal age, education, current employment status, wealth quintile, distance to the facility, complications during pregnancy, received ≥4 ANC visits, complications during delivery, mode of delivery (only for model 3 & 4), hard to reach areas, sample weight and clustering |
| Model 2 | No prelacteal feeding | | |
| Model 3 | Early initiation of breastfeeding (EIBF) | Delivery by medically trained provider (reference untrained provider) | |
| Model 4 | No prelacteal feeding | | |
| Model 5 | Early initiation of breastfeeding (EIBF) | Among facility births only: private sector delivery (reference public/ NGO sector delivery) | |
| Model 6 | No prelacteal feeding | | |
| Model 7 | Early initiation of breastfeeding (EIBF) | Cesarean section delivery (reference vaginal deliveries) | |
| Model 8 | No prelacteal feeding | | |

fits were tested using the Akaike information criterion (AIC) (based on log-likelihood) and the Bayesian information criterion (BIC) (based on deviance). Both measures of the model fits account for the number of parameters compared across the models. Smaller values were generally considered as better model fits.

## Ethics

The study was approved by the Research Review Committee (RRC) and Ethical Review Committee (ERC) of the Institutional Review Board (IRB) of icddr,b (protocol number PR#14044). Informed written consent in the local language was taken from the study participants before data collection. For illiterate participants, thumb impression was taken after explaining the details of the study objectives, risk, benefits, voluntary nature of participation and future use of data.

## Results

Among 2721 women included in this study, 60.7% were aged between 20 and 29 years. The percentage of women obtaining secondary or higher-level education was 48.8%. The majority of the women were unemployed during the time of the survey. More than half of the women had a health facility within 1 kilometer of their residence and around 26.3% of women received four or more ANCs during their last pregnancy. One-third of the sampled women had delivered in a facility, out of which 45.5% delivered in the private sector (Table 2). Around one third delivery of all delivery was conducted by medically trained provider.

The distribution of cesarean section and normal deliveries are displayed in Fig 2. Overall, the prevalence of cesarean section was 13% among all births and 46.7% of the facility deliveries were cesarean section deliveries. In contrast, 74.6% of the private sector deliveries were cesarean section deliveries.

### Distribution of favorable feeding practices

The distribution of EIBF and no prelacteal feeding by place of birth, delivery sector, and mode of delivery is tabulated in Table 3. Overall, 69.6% (95% CI: 67.8–71.3) women initiated breastfeeding within one hour of birth (EIBF) and 73.9% (95% CI: 72.3–75.6) newborns received only breast milk in the first three days of birth (no prelacteal feeding).

**Table 2. Descriptive analysis of women who had a live birth outcome in the 12 months preceding the survey (N = 2721) from four hard to reach areas of Bangladesh, 2017.**

| Variables | Categories | Frequency (N = 2721) | Percentages (%)* |
|---|---|---|---|
| Maternal age | 15–19 | 398 | 14.63 |
| | 20–29 | 1653 | 60.73 |
| | 30–39 | 670 | 24.64 |
| Educational level | No formal education | 433 | 15.89 |
| | Primary complete or below (1–5) | 960 | 35.29 |
| | Secondary and above (≥6) | 1328 | 48.82 |
| Current employment status | Unemployed | 2551 | 93.74 |
| | Employed | 170 | 6.26 |
| Religion | Muslim | 2439 | 89.65 |
| | Hindu /others | 282 | 10.35 |
| Wealth Quintiles | Lowest | 588 | 21.62 |
| | Second | 569 | 20.91 |
| | Middle | 546 | 20.05 |
| | Fourth | 516 | 18.95 |
| | Highest | 502 | 18.47 |
| Distance to the nearest health facility | Less than 1 km | 1448 | 54.31 |
| | 1–5 km | 1113 | 40.90 |
| | More than 5 km | 130 | 4.79 |
| Antenatal care(ANC) | No ANC | 712 | 26.16 |
| | 1 visit | 430 | 15.80 |
| | 2 visits | 416 | 15.28 |
| | 3 visits | 448 | 16.48 |
| | ≥4 visits | 715 | 26.28 |
| Complication during pregnancy | No | 2157 | 79.28 |
| | Yes | 564 | 20.72 |
| Complication during delivery | No | 1965 | 72.21 |
| | Yes | 756 | 27.79 |
| Place of delivery | Homebirth | 1984 | 71.60 |
| | Facility birth | 773 | 28.40 |
| Delivery Sector (among facility deliveries only) (N = 773) | Public sector | 331 | 42.80 |
| | Private sector | 351 | 45.45 |
| | NGO sector | 91 | 11.75 |
| Delivery by medically trained provider (N = 2721) | No | 1921 | 70.61 |
| | Yes | 800 | 29.39 |
| Hard to reach areas | Char | 727 | 26.7 |
| | Hilly | 733 | 26.9 |
| | Haor | 757 | 27.8 |
| | Coastal | 504 | 18.5 |

* Weighted by survey weight and population size of hard to reach areas.

## Home vs facility deliveries

EIBF was lower among women who had delivered in the facilities compared to the women delivering at home; 63.0% (95% CI: 59.5–66.4) vs. 72.2% (95% CI: 70.1–74.2), respectively. Contrastingly, the prevalence of no prelacteal feedings among home and facility birth was 73.2% and 75.8%, respectively (Table 3). Model 1, looking at the unadjusted association

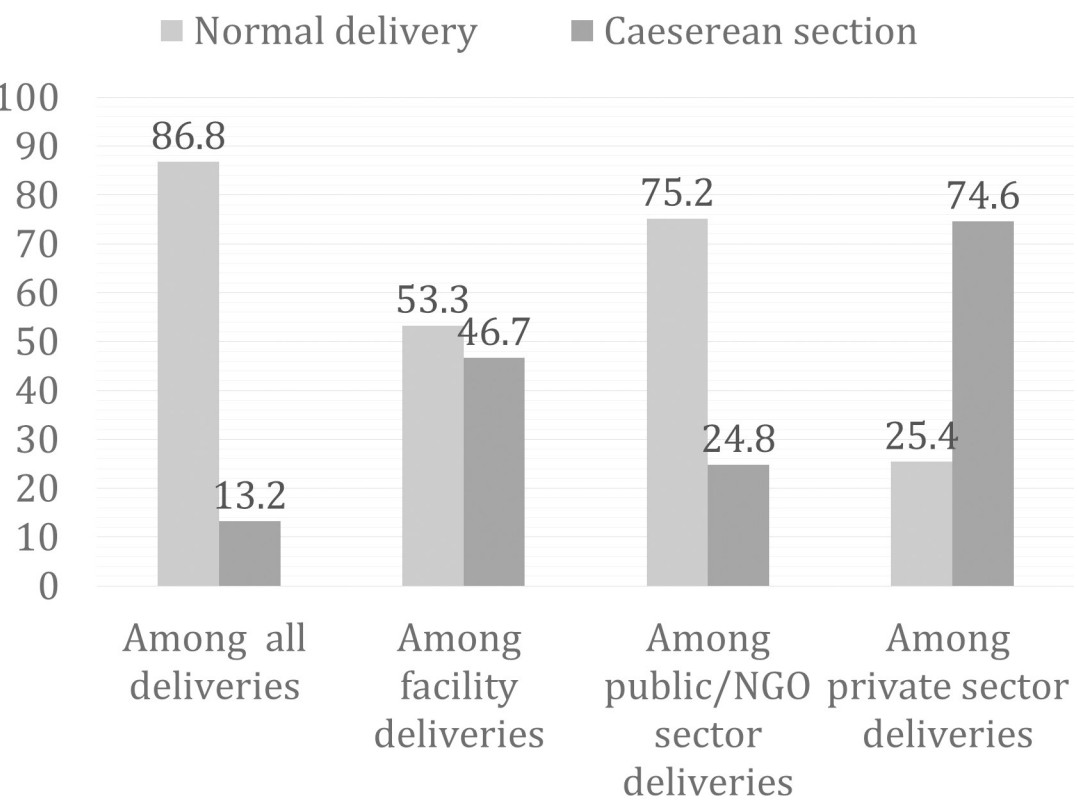

**Fig 2. Distribution of mode of delivery by place of delivery and delivery sector in four hard to reach areas of Bangladesh, 2017.**

between place of delivery and EIBF, found a significant association. After adjusting for potential confounders, we found that women delivering in the facilities had lower odds of EIBF (aOR = 0.51; 95% CI: 0.35–0.75, P<0.001) compared to women having home deliveries. On the other hand, Model 4 (unadjusted association of place of delivery and no prelacteal feeding) showed no significant association (Table 4).

**Table 3. Distribution of favorable feeding practices by place of delivery, delivery sector and mode of delivery among women who had a live birth outcome in past 12 months of the survey (N = 2721) in four hard to reach areas of Bangladesh, 2017.**

| Independent | | Dependent | | | |
|---|---|---|---|---|---|
| Variables | Categories | Early initiation of breastfeeding | | No pre-lacteal feeding | |
| | | Frequency | Percentages* (95% CI) | Frequency | Percentages* (95% CI) |
| Overall | | 1893 | 69.57 (67.80–71.29) | 2012 | 73.94 (72.25–75.58) |
| Place of delivery | Home birth | 1406 | 72.17 (70.12–74.15) | 1426 | 73.20 (71.17–75.15) |
| | Facility births | 487 | 63.00 (59.48–66.41) | 586 | 75.80 (72.62–78.78) |
| Delivery by medically trained provider | No | 1375 | 71.55 (69.45–73.53) | 1421 | 73.95 (71.89–75.87) |
| | Yes | 518 | 64.85 (61.40–68.14) | 591 | 73.98 (70.68–76.88) |
| Delivery Sector (among facility deliveries only) | Public/NGO sector | 292 | 69.19 (64.54–73.56) | 340 | 80.56 (76.46–84.23) |
| | Private sector | 195 | 55.55 (50.19–60.83) | 246 | 70.08 (64.99–74.83) |
| Mode of delivery | Vaginal delivery | 1708 | 72.37 (70.52–74.16) | 1769 | 74.95 (73.15–76.69) |
| | Cesarean section | 185 | 51.24 (45.96–56.51) | 243 | 67.31 (62.20–72.13) |

* Weighted by survey weight and population size of hard to reach areas.

## Medically trained provider Vs non medically trained provider

The reported prevalence of EIBF practices was lower among mothers whose delivery was conducted by medically trained provider at 64.9% (95% CI: 61.4–68.1) compared to the mothers who was delivered by non-medically trained provider at 71.6% (95% CI: 69.5–73.5) (Table 3). Our bivariate analysis showed that delivery by medically trained provider reduces the odds of EIBF by 52%, however no association was observed in adjusted analysis (Table 4). We did not find any association between no pre-lacteal feeding and delivery by medically trained provider.

**Table 4. Unadjusted and adjusted odds ratios (ORs) from generalized linear models (GLMs) (1–6).**

| Model | Exposure | Outcome | | Bivariate model[1] | Adjusted model[2] |
|---|---|---|---|---|---|
| 1 | Facility birth (reference home birth) | Early initiation of breastfeeding | OR | 0.51 | 0.51 |
| | | | P-Value | <0.000 | <0.001 |
| | | | 95% CI | 0.35–0.74 | 0.35–0.75 |
| | | | n | 2721 | 2721 |
| 2 | | No pre-lacteal feeding | OR | 1.31 | |
| | | | P-Value | 0.153 | |
| | | | 95% CI | 0.90–1.91 | |
| | | | n | 2721 | |
| 3 | Delivery by medically trained provider (Reference delivery by untrained provider) | Early initiation of breastfeeding (EIBF) | OR | 0.48 | 0.78 |
| | | | P-Value | <0.000 | 0.412 |
| | | | 95% CI | 0.33–0.71 | 0.44–1.40 |
| | | | n | 2721 | 2721 |
| 4 | | No pre-lacteal feeding | OR | 1.10 | |
| | | | P-Value | 0.531 | |
| | | | 95% CI | 0.81–1.49 | |
| | | | n | 2721 | |
| 5 | Private sector (reference public /NGO sector) | Early initiation of breastfeeding (EIBF) | OR | 0.53 | 1.04 |
| | | | P-Value | <0.001 | 0.851 |
| | | | 95% CI | 0.37–0.77 | 0.64–1.68 |
| | | | | 773 | 773 |
| 6 | | No pre-lacteal feeding | OR | 0.54 | 0.68 |
| | | | P-Value | <0.05 | 0.144 |
| | | | 95% CI | 0.35–0.83 | 0.40–1.14 |
| | | | n | 773 | 773 |
| 7 | Cesarean section(reference vaginal delivery) | Early initiation of breastfeeding | OR | 0.22 | 0.20 |
| | | | P-Value | <0.000 | <0.000 |
| | | | 95% CI | 0.14–0.34 | 0.12–0.35 |
| | | | n | 2721 | 2721 |
| 8 | | No pre-lacteal feeding | OR | 0.73 | |
| | | | P-Value | 0.176 | |
| | | | 95% CI | 0.45–1.15 | |
| | | | n | 2721 | |

[1] Adjusted for hard to reach areas and clustering.

[2] Adjusted for maternal age, education, current employment status, wealth quintile, distance to the facility, complications during pregnancy, received ≥4 ANC visits, complications during delivery, mode of delivery (model 3,4,5& 6), hard to reach areas, sample weight and clustering.

### Public/NGO vs private sector deliveries

Among the women delivering in the public facilities, 69.2% reported EIBF and 80.6% reported no prelacteal feeding. Prevalence of EIBF and no prelacteal feeding was lower among women delivering in the private sector compared to the public sector (Table 3). The detailed results of the unadjusted and adjusted associations between the delivery sector and favorable feeding practices (EIBF and no pre-lacteal feeding) could be found in Table 4. Model 5, showing the unadjusted association between the delivery sector and EIBF, showed lower odds of EIBF among women delivering in the private sector compared to the women who had delivered in the public sector; OR 0.53 (95% CI: 0.37–0.77; P-value <0.001). After adjusting for all potential confounders, including mode of delivery, we found no significant association between the delivery sector and EIBF; aOR 1.04 (95% CI: 0.64–1.68; P-value 0.851). Similarly, the adjusted association between the delivery sector and no pre-lacteal feeding in Model 6 was statistically insignificant.

### Vaginal deliveries vs. cesarean sections

Results suggest that 72.4% of women having vaginal delivery had initiated breastfeeding within one hour of birth, while around half of the women having cesarean section reported EIBF (51.2%) (Table 3). No prelacteal feeding was also lower amongst women having a cesarean section (67.3%) compared to women having a vaginal birth (74.9%). Model 7 indicated that women having cesarean section had lower odds of EIBF after adjusting for potential confounders (aOR = 0.20, 95% CI: 0.12–0.35, P<0.000) compared to women having vaginal births (Table 4). No significant association was found between cesarean section deliveries and no pre-lacteal feeding (Model 8).

## Discussions

Our findings provide critical evidence on the association of childbirth location and mode of delivery with favorable breastfeeding practices in four HtR areas of Bangladesh. Our results suggest that the women having ceasarian section deliveries were less likely to engage in EIBF and prelacteal feeding practices. Therefore, with the growing changes in the dynamics of facility deliveries in Bangladesh (as evidenced from the trend assessments of BDHS 2014 and 2017), the EIBF practices may further go down in future. This study highlights the need to ensure favorable breastfeeding conditions for women delivering at facilities. Furthermore, this study also indicates that there could be potential barriers that is precluding the adoption of favorable breastfeeding practices at facility studies. The study is unique as it employs the most recent datasets (following the year 2014) and presents the recent breastfeeding practices in HtR areas.

We found around 69.6% of newborns started breastfeeding within one hour of birth and 73.9% were given only breastmilk in the first three days (no prelacteal feeding). Our findings are consistent with the prevalence of EIBF in rural areas of Bangladesh in 2017[22]. Our secondary analysis of BDHS 2014 data showed that the prevalence of EIBF was 68.9% among women living in rural areas and was higher than women residing in urban areas [17].

For women who had delivered in the facilities, the adjusted odds of initiating breastfeeding within one hour of birth were 49% lower than the women had delivered at home. Our finding is similar to other studies conducted in similar settings [19, 20, 26]. We found that the separation of newborns from mothers to perform essential interventions after delivery is a common practice in the facilities and is the major barrier to early initiation of breastfeeding [20, 27, 28]. Additionally, lack of privacy in the delivery room and low prevalence of skin to skin contact within one hour of delivery significantly reduces EIBF practices in the facilities [26]. Initiation

of skin to skin contact within one hour of births is a WHO-recommended essential newborn care practice, which prevents hypothermia among newborns and improves EIBF practices [29]. The experience from Alive and Thrive programme from implementing the Early Essential Newborn Care Practices (EENCP) intervention package at 102 hospitals in Vietnam found that training and supportive supervision of health care providers can help improve EIBF practices after vaginal deliveries [30]. Implementation of quality of care improvement initiatives, as recommended by WHO, can also enhance EIBF practices in the facilities by ensuring positive pregnancy experience among mothers and creating enabling environments for EIBF practices [31].

Cesarean section deliveries are inversely associated with favorable early breastfeeding practices [2, 11, 19, 32–35]. In our analysis (Model7), we found that the mothers undergoing cesarean section had 80% lower odds of initiating early breastfeeding than mothers having vaginal deliveries. Complicated pregnancies usually result in cesarean deliveries where mothers may not be physically fit to start breastfeeding within one hour of birth [36, 37]. However, we adjusted for complications during pregnancy and delivery and still found a significant reduction of EIBF for deliveries with a cesarean section. Furthermore, evidence suggests that routine postnatal care and separation of the newborn after cesarean section delays early initiation of breastfeeding by interrupting the natural bonding between mother and infant. This alters the release of essential hormones (oxytocin & prolactin), which are necessary for the steady production and supply of breast milk [28, 38, 39]. Rowe et al. emphasized not to separate mothers and newborns during the immediate postpartum period after cesarean deliveries [35]. Their study provided several recommendations for improving early breastfeeding practices, which include keeping newborns in the postoperative room with the mother, initiating skin to skin contact, delaying neonatal monitoring for healthy newborns in the first hour of birth and ensuring the presence of support staff or birth companion in the postnatal room. The absence of support staff after cesarean deliveries is recognized as a barrier to EIBF [30]. WHO recommends the presence of a birth companion during labor and the immediate postpartum period [31]. Evidence suggests that a birth companion's presence can improve early initiation of breastfeeding practices after cesarean delivery [37, 40].

WHO and UNICEF recommend every country adopt "Baby-Friendly Hospital Initiatives" to improve early breastfeeding practices in the facilities [41]. It includes the training of the health care providers to create an enabling environment for improving favorable breastfeeding practices [28]. Systematic evaluation of the "Baby-Friendly Hospital Initiative" showed significant improvement in EIBF in the practicing facilities compared to the non-practicing facilities [35, 42–46]. Although the Government of Bangladesh (GoB) recently started implementing the "Baby-Friendly Hospital Initiative" in the facilities, ensuring the availability and accessibility of essential health services in HtR areas is a big challenge. The shortage of skilled health workforce is one of the major barriers for ensuring health service availability for geographically inaccessible areas with poor road networks, low literacy rate and high level of poverty [23, 47–49].

Our study results show that the coverage of facility delivery in four HtR areas of Bangladesh was 28.4%, considerably lower than the national average of 50% [22]. This finding is not surprising considering the nature of HtR areas and preference for home deliveries in rural areas of Bangladesh [50, 51]. In this regard, a differential or targeted programming to improve health service availability and accessibility in HtR areas could be a potential solution. Hossain et al. 2020 explained the possibility of filling up human resource gaps in remote areas of Bangladesh by deploying private community-based skilled birth attendants [52]. The notable success of Bangladesh in reducing maternal and child mortality was attributed to the innovative community-based approaches and public-private partnerships [53]. Significant improvement

in early breastfeeding practices was observed following the training of the traditional birth attendants [54].

Another interesting finding of our study is that about 50% of the women having facility deliveries were from the private sector. We found a significant negative association between EIBF and private sector deliveries, which became insignificant after adjusting for potential confounders, especially the mode of delivery. This could be due to the high rates of cesarean deliveries in the private sector of hard to reach areas (74%). In 2017 and at the national level, about 84% of deliveries in the private sector were cesarean sections [22]. This finding calls for urgent measures to regulate unnecessary cesarean sections in private facilities of HtR areas. Licensing of the private facilities, regular audit by quality improvement team, monitoring of caesarean births using Robson's 10 group classification, second opinion for caesarean births were recommended for reduction of unnecessary caesarean sections in the profit driven private facilities [55–57].

The finding from this study should be interpreted considering the following limitations. First, we used the self-reported data from mothers to understand when breastfeeding was initiated after the birth. This information was later used for defining the EIBF. The recall bias should be considered while interpreting the results from self-reported data. However, to minimize the effect of recall bias, we interviewed mothers who have their last birth outcome within the past 12 months of the survey, thus reducing the recall period. Second, the use of cross-sectional data did not allow us to establish a temporal relationship between the exposure and outcome variables. Further study is needed to establish causality through cohort follow up. Third, we employed the generalized linear modeling technique to assess the association between the exposure and outcome variables. In real life, the relationships between the variables are not so straightforwardly linear. However, given our research objectives, we found the GLM modeling approach to be the most suitable to analyze the associations between our target variables.

Despite the limitations, this study took up the challenge of understanding the dynamics of early breastfeeding practices in HtR areas. Our study presents novel findings that are highly contextual with regards to the recent changes in the key maternal and newborn health indicators of Bangladesh. The findings are expected to assist policymakers in undertaking targeted intervention strategies to reduce newborn mortality in areas with poor accessibility to healthcare.

## Conclusion

Early initiation of breastfeeding and no prelacteal feeding are essential interventions for improving newborn survival, growth and development. Our paper highlights critical associations between child birthplace and mode of delivery with favorable early breastfeeding practices in four hard to reach areas of Bangladesh. Implementation and scale-up of "Baby-Friendly Hospital Initiatives" could help improve EIBF practices after facility birth and cesarean deliveries. Given the challenges of ensuring essential health services in the hard to reach areas, differential targeted programming could be one of the potential solutions. Further studies are needed to design and test the efficacy of differential package of interventions for improving early breastfeeding practices after facility births and cesarean deliveries in hard to reach areas of Bangladesh.

## Acknowledgments

We acknowledge with gratitude the commitment of Swedish International Development Cooperation Agency (Sida) to their research efforts in Bangladesh. We are also grateful to the Government of the People's Republic of Bangladesh, Global Affairs Canada (GAC), Swedish

International Development Cooperation Agency (Sida), and the Department for International Development (UKAid) for providing core/unrestricted support to icddr,b. We are thankful to our study participants and our data collection and management team for their contribution to this study.

## Author Contributions

**Conceptualization:** Nazia Binte Ali.

**Data curation:** Nazia Binte Ali.

**Formal analysis:** Nazia Binte Ali, Farhana Karim.

**Funding acquisition:** Farhana Karim, S. K. Masum Billah, Dewan M. D. Emdadul Hoque, Abdullah Nurus Salam Khan, Mohammad Mehedi Hasan, Mohiuddin Ahsanul Kabir Chowdhury.

**Investigation:** Nazia Binte Ali, Farhana Karim, S. K. Masum Billah, Dewan M. D. Emdadul Hoque, Abdullah Nurus Salam Khan, Mohammad Mehedi Hasan, Mohiuddin Ahsanul Kabir Chowdhury.

**Methodology:** Nazia Binte Ali, Farhana Karim, S. K. Masum Billah, Dewan M. D. Emdadul Hoque, Abdullah Nurus Salam Khan, Mohammad Mehedi Hasan, Mohiuddin Ahsanul Kabir Chowdhury.

**Project administration:** Nazia Binte Ali, Farhana Karim, Abdullah Nurus Salam Khan, Mohammad Mehedi Hasan, Mohiuddin Ahsanul Kabir Chowdhury.

**Resources:** Mohammad Mehedi Hasan, Mohiuddin Ahsanul Kabir Chowdhury.

**Software:** Mohammad Mehedi Hasan, Mohiuddin Ahsanul Kabir Chowdhury.

**Supervision:** Farhana Karim, Mohammad Mehedi Hasan, Mohiuddin Ahsanul Kabir Chowdhury.

**Validation:** Nazia Binte Ali, Mohiuddin Ahsanul Kabir Chowdhury.

**Visualization:** Nazia Binte Ali, Mohiuddin Ahsanul Kabir Chowdhury.

**Writing – original draft:** Nazia Binte Ali, Sonjida Mesket Simi, Mohiuddin Ahsanul Kabir Chowdhury.

**Writing – review & editing:** Nazia Binte Ali, Farhana Karim, S. K. Masum Billah, Dewan M. D. Emdadul Hoque, Abdullah Nurus Salam Khan, Mohammad Mehedi Hasan, Shams E. L. Arifeen, Mohiuddin Ahsanul Kabir Chowdhury.

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
