## [Decision Letter · Decision Letter 0]

5 Oct 2020

PONE-D-20-13424

Are childbirth location and mode of delivery associated with favorable early breastfeeding practices in hard to reach areas of Bangladesh?

PLOS ONE

Dear Dr. Ali,

Thank you for submitting your manuscript to PLOS ONE. After careful consideration, we feel that it has merit but does not fully meet PLOS ONE’s publication criteria as it currently stands. Therefore, we invite you to submit a revised version of the manuscript that addresses the points raised during the review process.

I congratulate the authors to an interesting and well written paper. As academic editor, given the difficulties in finding adequate reviewers, I read the manuscript from a review perspective. I have only minor comments in addition to the other review and recommend a minor revision. Please see comments below.

*PLOS ONE* requires data sharing and deposition in public databases - please comment on this requirement.

We look forward to receiving your revised manuscript.

Kind regards,

Julia Dratva

Academic Editor

PLOS ONE

Journal Requirements:

2.Thank you for including your ethics statement:  "The study was approved by the Institutional Review Board (IRB) of icddr,b(protocol number PR#14044). Informed written consent in the local language was taken from the study participants before data collection. "

3.We suggest you thoroughly copyedit your manuscript for language usage, spelling, and grammar. If you do not know anyone who can help you do this, you may wish to consider employing a professional scientific editing service.  

4.We note that you have indicated that data from this study are available upon request. PLOS only allows data to be available upon request if there are legal or ethical restrictions on sharing data publicly. For more information on unacceptable data access restrictions, please see http://journals.plos.org/plosone/s/data-availability#loc-unacceptable-data-access-restrictions.

Additional Editor Comments (if provided):

I congratulate the authors to an interesting and well written paper. As academic editor, given the difficulties in finding adequate reviewers, I read the manuscript from a review perspective. I have only minor comments in addition to the other review and recommend a minor revision.

Abstract:

Unclear what respectively alludes to, I guess birth location. However, as you present birth location in the methods: place of birth (home vs facility), the sentence in line 132, could be understood that 69.9% refers to home births. Need to be clearer to avoid misunderstanding.

Line 32 The prevalence of EIBF and no prelacteal feeding were 69.6%(95% CI:67.8-71.3) and 73.9% 33 (95% CI:72.3-75.6) respectively

Methods:

Line 134: We did not categorize home births by type of providers.

I would be interested to know, howe many types and which providers there are in the area.

Would this information be of relevance for the interpretation of the results, may one provder type drive the result? Please comment on this.

Reviewers' comments:

Reviewer's Responses to Questions

**Comments to the Author**

1. Is the manuscript technically sound, and do the data support the conclusions?

Reviewer #1: Yes

2. Has the statistical analysis been performed appropriately and rigorously? 

Reviewer #1: Yes

3. Have the authors made all data underlying the findings in their manuscript fully available?

Reviewer #1: No

4. Is the manuscript presented in an intelligible fashion and written in standard English?

Reviewer #1: Yes

5. Review Comments to the Author

Reviewer #1: Thank you! For coming up with this an interesting topic. I am happy with your manuscript which provides important findings on factors related with breastfeeding practices. With this I have the following comments and questions

In Background page 5 Line 78. Can you show us the pattern in change of health seeking behaviors? Unless it becomes personal suggestion nor it’s supported with evidences.

In Method section Page 5 Line 90. icddr,what does this mean ?

In Method section Page 5 Line 91. How did you decided that the areas were hard to reach? It is better to describe about the study areas basically with respect to health profiles to better understand the area.

In Method section Page 6 Line 100. Is the questionnaire validated/reliable? Did you measure Cronbach's alpha and test the reliability/validity of the tool with your sample? Have you conducted pretest before you conduct the actual study?

In discussion section page 17 line 270. Evidences suggest that during facility deliveries mothers born by C/S had low rate of EIBF practices. SO what do you think other essential intervention that prevent EIBF practices? Identifying this intervention were important to recommend for the health care professionals in order to increase EIBF practices.

6. PLOS authors have the option to publish the peer review history of their article (what does this mean?). If published, this will include your full peer review and any attached files.

Reviewer #1: No

---

## [Author Response · Author response to Decision Letter 0]

15 Oct 2020

Additional Editor Comments (if provided):

Abstract:

Unclear what respectively alludes to, I guess birth location. However, as you present birth location in the methods: place of birth (home vs facility), the sentence in line 132, could be understood that 69.9% refers to home births. Need to be clearer to avoid misunderstanding.

Line 32 The prevalence of EIBF and no prelacteal feeding were 69.6%(95% CI:67.8-71.3) and 73.9% 33 (95% CI:72.3-75.6) respectively

Response: Thanks for the suggestions. We have updated the relevant section (line 33-35) as below 

“The prevalence of EIBF practices were 69.6%(95% CI:67.8-71.3); 72.2%(95% CI:67.8-71.3) among home births Vs 63.0%(95% CI:59.5%-66.4%) among facility birtsh. Around 73.9% (95% CI:72.3-75.6) mother’s in the study areas reported no-prelacteal feeding”

Methods:

Line 134: We did not categorize home births by type of providers.

I would be interested to know, howe many types and which providers there are in the area.

Would this information be of relevance for the interpretation of the results, may one provder type drive the result? Please comment on this.

Response: Thanks for the suggestions. We added types of birth attendant table 2. We did cross tabulation to check whether the prevalence of EIBF and no pre-lacteal feeding varies by provider type or not (table 3). We have added additional model in table 4 and description in 239-245. Our unadjusted analysis showed that the odds of having EIBF decreases if the delivery was conducted by medically trained provider compared to the delivery by non-medically trained provider however we did not find any association in adjusted model(3). 

Reviewers' comments:

Reviewer's Responses to Questions

Reviewer #1: Thank you! For coming up with this an interesting topic. I am happy with your manuscript which provides important findings on factors related with breastfeeding practices. With this I have the following comments and questions

In Background page 5 Line 78. Can you show us the pattern in change of health seeking behaviors? Unless it becomes personal suggestion nor it’s supported with evidences.

Response: 

Thanks for the query. BDHS 2014 and BDHS 2017 depicts that the coverage of facility delivery increased from 37% in 2014 to 50% in 2017. In reference to this data we mentioned the change in care seeking in delivery services. We have rephrased the section for better understanding as below 

“First, while comparing the trends from Bangladesh Demographic and Health Survey (BDHS) in 2014 and 2017, we observed some major shifts in the coverage of facility deliveries (37% vs. 50%), deliveries at private sector (22% vs. 32%) and cesarean section rates (23% vs. 33%)(21, 22). Similar improvements were also observed in the prevalence of EIBF, which increased from 51% to 69% between 2014 and 2017(21, 22). Due to these recent changes, it is essential to reexamine the associations between the place of birth and mode of delivery with EIBF and no preleacteal feeding practices.”

In Method section Page 5 Line 90. icddr, what does this mean ?

Response: This is a typo. We have corrected in lines as below

“A cross-sectional study was conducted by Maternal and Child Health Division of International Center for Diarrheal Disease Research, Bangladesh (icddr,b) in four hard to reach (HtR) areas (Char, Hilly, Haor, and Coastal) of Bangladesh in 2017.”

In Method section Page 5 Line 91. How did you decided that the areas were hard to reach? It is better to describe about the study areas basically with respect to health profiles to better understand the area.

Response:

Thanks for the query. Hard to reach areas are defined by Government of Bangladesh. Government of Bangladesh used six indicators for defining hard to areas. We used government lists for selecting study areas and added reference. The texts are updated as below

“ The Government of Bangladesh (GoB) determines these lists of HtR areas and uses six indicators for defining HtR sub-districts; land type, availability of water, access to pure drinking water, sanitation facilities, poverty level, and child mortality rate (23). ”

In Method section Page 6 Line 100. Is the questionnaire validated/reliable? Did you measure Cronbach's alpha and test the reliability/validity of the tool with your sample? Have you conducted pretest before you conduct the actual study?

Response: We used household and women’s module of Bangladesh Demographic and Health Survey (BDHS) 2014 questionnaire for data collection. 

“Household and women module of BDHS questionnaire were based on the model questionnaires developed for the international DHS-6 Program, adapted to the situation and needs in Bangladesh and taking into account the content of the instruments employed in prior DHS surveys in Bangladesh. BDHS questionnaires were adapted for use in Bangladesh during a series of meetings with a Technical Working Group (TWG). Draft questionnaires were then circulated to other interested groups and were reviewed by the BDHS Stakeholder Advisory Committee (SAC).” 

As we directly used BDHS modules without any alteration, we did not test validity or reliability of our tools. We used reference of BDHS in our manuscript. 

In discussion section page 17 line 270. Evidences suggest that during facility deliveries mothers born by C/S had low rate of EIBF practices. SO what do you think other essential intervention that prevent EIBF practices? Identifying this intervention were important to recommend for the health care professionals in order to increase EIBF practices.

Response: Thanks for suggestion. We have included interventions recommended for reduction of unnecessary caesarean sections in 351-354 as below;

“Licensing of the private facilities, regular audit by quality improvement team, monitoring of caesarean births using Robson’s 10 group classification, second opinion for caesarean births were recommended for reduction of unnecessary caesarean sections in the profit driven private facilities(55-57).”

---

## [Editor Report · Decision Letter 1]

28 Oct 2020

Are childbirth location and mode of delivery associated with favorable early breastfeeding practices in hard to reach areas of Bangladesh?

PONE-D-20-13424R1

Dear Dr. Ali,

We’re pleased to inform you that your manuscript has been judged scientifically suitable for publication and will be formally accepted for publication once it meets all outstanding technical requirements.

Kind regards,

Julia Dratva

Academic Editor

PLOS ONE

Additional Editor Comments (optional):

Dear authors

thank you for the revision of the paper, which was clear and provided the requested clarification and additional information for the readers.

Editing of the paper inproved the readability and flow.

I suggest to accept your paper as revised.
---

## [Editor Report · Acceptance letter]

5 Nov 2020

PONE-D-20-13424R1 

Are childbirth location and mode of delivery associated with favorable early breastfeeding practices in hard to reach areas of Bangladesh?  

Dear Dr. Ali:

I'm pleased to inform you that your manuscript has been deemed suitable for publication in PLOS ONE. Congratulations! Your manuscript is now with our production department. 

Kind regards, 

on behalf of

Dr. Julia Dratva 

Academic Editor

PLOS ONE